# ALS’ Perfect Storm: *C9orf72*-Associated Toxic Dipeptide Repeats as Potential Multipotent Disruptors of Protein Homeostasis

**DOI:** 10.3390/cells13020178

**Published:** 2024-01-17

**Authors:** Paulien H. Smeele, Giuliana Cesare, Thomas Vaccari

**Affiliations:** Department of Biosciences, University of Milan, Via Celoria 26, 20133 Milan, Italy

**Keywords:** protein homeostasis, ALS, FTD, *C9orf72*, dipeptide repeats, autophagy, ER stress, proteasome, lysosome

## Abstract

Protein homeostasis is essential for neuron longevity, requiring a balanced regulation between protein synthesis and degradation. The clearance of misfolded and aggregated proteins, mediated by autophagy and the ubiquitin–proteasome systems, maintains protein homeostasis in neurons, which are post-mitotic and thus cannot use cell division to diminish the burden of misfolded proteins. When protein clearance pathways are overwhelmed or otherwise disrupted, the accumulation of misfolded or aggregated proteins can lead to the activation of ER stress and the formation of stress granules, which predominantly attempt to restore the homeostasis by suppressing global protein translation. Alterations in these processes have been widely reported among studies investigating the toxic function of dipeptide repeats (DPRs) produced by G4C2 expansion in the C9orf72 gene of patients with amyotrophic lateral sclerosis (ALS) and frontotemporal dementia (FTD). In this review, we outline the modalities of DPR-induced disruptions in protein homeostasis observed in a wide range of models of C9orf72-linked ALS/FTD. We also discuss the relative importance of each DPR for toxicity, possible synergies between DPRs, and discuss the possible functional relevance of DPR aggregation to disease pathogenesis. Finally, we highlight the interdependencies of the observed effects and reflect on the importance of feedback and feedforward mechanisms in their contribution to disease progression. A better understanding of DPR-associated disease pathogenesis discussed in this review might shed light on disease vulnerabilities that may be amenable with therapeutic interventions.

## 1. Introduction

It is now well established that aging is associated with a progressive dysregulation in protein homeostasis [1,2]. Consistent with this, a key hallmark of many neurodegenerative diseases is the accumulation of toxic protein aggregates, ultimately leading to progressive loss of neuronal structure and function. This is the case for amyotrophic lateral sclerosis (ALS) and frontotemporal dementia (FTD), two incurable diseases that share overlapping clinical manifestations, pathogenic mechanisms, and genetic risk factors [3,4]. In fact, many patients of both diseases have been found to carry a G4C2 hexanucleotide repeat expansion (HRE) in a non-coding region of the *C9orf72* gene [5,6].

While, to date, most patients affected by C9orf72-linked ALS/FTD (C9-ALS/FTD) harbor between 100 and 1000 G4C2 repeats, the exact disease-causing threshold and how repeat length might alter disease progression, remains to be determined. Indeed, several studies have shown that expansions of around 30 repeats could be sufficient to be disease-causing [7,8,9]. In particular, the expansion is thought to cause disease via two mutually inclusive mechanisms: (1) a loss of function of the *C9orf72* gene [10] and (2) a toxic gain of function driven by the HRE itself.

The relative importance of each of the two mechanisms represents a controversial, unanswered question at the core of the C9-ALS/FTD field. Indeed, since the discovery C9orf72-linked ALS/FTD, many conflicting results have been published regarding the relative importance of each of the two mechanisms. The lack of a clear correlation between repeat length, for example, appears to argue against the toxic gain-of-function as being a major driver of disease pathology. In line with this a recent phase I clinical trial (BIIB078) of an antisense oligonucleotide (ASO), targeting G4C2 repeat RNA and thus also DPR generation, failed to show any clinical benefits. It is unclear, however, whether the ASO administration indeed significantly lowered the levels of repeat RNAs and DPRs. Arguing instead against the loss-of-function hypothesis, patients homozygous for the *C9orf72* repeat expansion, and thus expressing less *C9orf72*, do not show a more serve phenotype than those with *C9orf72* haploinsufficiency [11]. In addition, *C9orf72* knockout mice do not show any ALS/FTD-associated neurodegenerative phenotypes [12]. Although this remains an important unanswered question to be addressed in the field, the two mechanisms likely act in coordination, and it is therefore important to continue investigating both hypotheses.

Interestingly, the overexpression of a sufficient number of G4C2 repeats per se (i.e., without modulation of *C9orf72* gene dosage) in vitro [13,14,15] as well as in mice [16,17], zebrafish [18,19], and *Drosophila melanogaster* [20,21] is sufficient to induce toxicity and neurodegeneration. These can be bidirectionally transcribed into repeat RNAs that subsequently aggregate, a general hallmark of non-coding repeat expansion diseases. Indeed, samples from patients with C9-ALS/FTD and patient-derived induced pluripotent stem cells (iPSCs) have been shown to contain nuclear, as well as some cytoplasmic, foci of aggregated RNA [18,20,22,23,24]. Further, these sense and anti-sense repeat RNAs can undergo non-canonical, repeat-associated non-AUG (RAN) translation to produce five distinct dipeptide repeats (DPRs): poly-PA, poly-PR, poly-GA, poly-GP, and poly-GR [25,26,27,28,29].

In this review, we focus on the role of DPRs in disrupting protein homeostasis pathways, and on recent evidence revealing how DPRs may contribute to multiple aspects of disease pathogenesis.

## 2. The Impact of C9-Associated Toxic Repeats on Protein Degradation Pathways

A key neuropathological hallmark in C9-ALS/FTD is the formation of star-shaped cytoplasmic DPR inclusions which are ubiquitin- and p62-positive [30,31], indicative of an involvement of the ubiquitin–proteasome system (UPS) and of macroautophagy (hereafter autophagy) in the targeted degradation of DPRs. Consistent with this, the pharmacological inhibition of either pathway leads to an increased accumulation and aggregation of distinct DPRs in cell culture [14,32]. However, several recent studies suggest that, while DPRs are targets of UPS- and autophagy-mediated clearance, they might also play a major role in disrupting both pathways (Figure 1), as discussed in detail in the next two chapters.

### 2.1. Poly-GA Inhibits Protein Degradation via the UPS

The UPS is the major cellular pathway responsible for the degradation and recycling of short-lived, soluble proteins, and indeed appears to be important for DPR degradation [14,32]. In addition to being possible targets of UPS-mediated degradation, DPRs in fact appear to inhibit UPS activity [13,14,15,33,34,35,36,37]. Interestingly, several mutations in genes involved in the UPS have been associated with both ALS and FTD [4], highlighting the UPS as a common point of vulnerability in both neurodegenerative diseases. While all DPRs, with the notable exception of poly-PA, have been associated with disrupted UPS, the underlying mechanisms remain unclear.

Several studies have shown that UPS factors co-localize with cytoplasmic poly-GA inclusions in cell culture [13,14,15,33,34], in animal models, and in the brains of patients with C9-ALS/FTD [35]. Pointing to a more direct impairment of the UPS system by accumulation poly-GA, it has recently been shown that the 26S proteasome, which is ultimately required to degrade ubiquitin-tagged proteasomal substrates, is sequestered into poly-GA aggregates in cultured neurons [33]. In situ structural analysis of neuronal poly-GA aggregates suggests that these may force the sequestered proteasomes to become stuck in a highly transient intermediate state, which is usually associated with substrate translocation. This may lead to stalled degradation of ubiquitinated substrates, which could explain observed reductions in proteasome activity [33].

The specific consequences of Impaired proteasome function in the context of C9-linked ALS/FTD remains to be further investigated. However, independent of C9-ALS/FTD, the inhibition of the 26S proteasome has also been shown to lead to the cytoplasmic mislocalization and aggregation of TAR DNA-binding protein 43 (TDP43) [38,39], a second aggregation-prone protein associated with the vast majority of familiar and sporadic forms of ALS. Indeed, some TDP-43 pathology has been observed in patients with C9-ALS/FTD [28,40,41], although it appears to occur after DPR pathology [42,43]. Whether proteasomal inhibition by poly-GA may be a contributing factor to TDP-43 pathology therefore warrants further investigation. Indeed, primitive evidence suggests that poly-GA aggregates are able to induce TDP-43 mislocalization in cell culture, and that this mislocalization is dependent on GA-induced inhibition of the proteasome [34]. Interestingly, proteasome inhibition was shown to occur in a cell-autonomous and non-cell-autonomous manner. Specifically, cytoplasmic TDP-43 was observed in rat primary hippocampal neurons cultured with cell supernatant from GA-transduced cells. Such TDP-43 mislocalization was eliminated when depleting the culture media of poly-GA with anti-GA antibodies. While is remains to be determined whether this is through cell-to-cell transmission, this mechanism may help to explain why DPR inclusions and TDP-43 pathology predominantly occur in distinct neurons within the brains of patients with C9-ALS/FTD [41].

### 2.2. C9-Associated Toxic Repeats Disrupt Autophagosome and Lysosome Biogenesis

In addition to the UPS, the autophagy pathway is another essential contributor to intracellular protein clearance, which predominantly targets insoluble, aggregated, and long-lived proteins. The autophagy pathway involves the recognition of ubiquitin-tagged proteins destined to clearance by adaptor proteins, such as p62/SQSTM1, promoting engulfment by the forming autophagosome, a specialized double-membrane organelle [44]. In neurons, autophagosomes are predominantly formed in distal regions of the axon and, as they undergo retrograde transport toward the soma, they mature and ultimately fuse with lysosomes, a process critical for neuronal longevity [45]. Importantly, lysosomes contain digestive enzymes, activated by a low pH, which eventually break down the autophagic cargo, allowing the degraded products to be recycled back to the cell [46].

Emerging evidence suggests that both repeat RNAs and DPRs may disrupt multiple steps of the autophagy pathway. The overexpression of G4C2 repeats in *Drosophila* motor neurons, for example, leads to an accumulation of *Drosophila* p62, accompanied by a reduction in the number of autophagosomal vesicles in both the soma and distal axons in vivo [47,48]. These data suggest that autophagy initiation, and specifically autophagosome formation, may be impaired. In line with this, G4C2 repeat expression results in reductions in the levels of mature autophagosomes in cell culture [49].

How could autophagy induction be affected by the expression of G4C2 repeats? One possibility is that reductions in autophagosome formation may be, at least in part, a result of disruptions in TFEB, the master transcriptional regulator of autophagy and lysosomal biogenesis (Figure 1C). Specifically, G4C2 toxicity appears to inhibit the nuclear import of TFEB in *Drosophila* as well as in cell culture [47,49]. Remarkably, dysfunction in nucleo-cytoplasmic transport and the nuclear pore complex (NPC) is emerging as a key contributor to disease pathogenesis [50]. This includes disruptions induced by repeat RNAs in *Drosophila* neurons and patient-derived iPSCs [51,52], as well as by DPRs. In fact, using *Xenopus laevis* oocytes as a model system, poly-PR has specifically been shown to bind to the central channel nuclear pore to induce a block in the transport of macromolecules between the nucleus and cytoplasm [53]. Additionally, both cytoplasmic poly-GA and poly-GR aggregates appear to sequester components of the NPC in the brains of DPR-expressing mice and patients with C9-ALS/FTD [35,54]. This includes the nucleoporin POM121, which when overexpressed leads to a rescue of TFEB nuclear localization and autophagy initiation in cell culture [49].

In addition to promoting autophagosome formation, TFEB is essential in mammals and in flies for the expression of genes required for lysosome biogenesis and function [55,56,57]. Thus, it is also possible that lysosomes might be defective in the presence of repeat RNAs and DPRs. Consistent with this possibility, G4C2 overexpression results in reduced cleavage and activation of a *Drosophila* cathepsin and might result in reduced lysosome acidification. Indeed, the overexpression of vacuolar ATPase (V-ATPase) genes, encoding the main proton pump required for lysosome acidification, appear to suppress G4C2-induced neurodegeneration in *Drosophila* [47]. Interestingly, the specific overexpression of only poly-GA in human cells has been shown to lead to the accumulation of mature autophagosomes, as well as p62 and ubiquitin, which is indicative of lysosomal impairment but not necessarily disruptions in autophagosome formation [58]. This discrepancy may be a result of the different system used, as well as the differences arising from the expression of a single DPR versus all DPRs simultaneously (e.g., G4C2 repeat overexpression).

Despite the accumulating evidence of the G4C2-induced impairment of autophagy and lysosomal biogenesis, it is important to note that *C9orf72*-deficient cells also display reduced levels of autophagy initiation and impaired lysosome biogenesis, leading to enhanced DPR accumulation and increased neurotoxicity, indicating that the product of *C9orf72* might play a role in physiologic activation of autophagy [59,60,61,62,63,64,65]. This is not surprising as C9orf72 is likely to act as an GDP/GTP exchange factor (GEF) for a number of RabGTPases regulating the early steps of autophagy and endocytic trafficking [66].

While reduced autophagy may thus represent an important contributor to C9-ALS/FTD pathogenesis, the consequences of reduced autophagy might extend beyond the impaired degradation of the DPRs themselves. Intriguingly, Marchi et al. recently proposed that reduced lysosome acidification may represent a mechanism by which endocytosed poly-GA aggregates can circumvent lysosomal degradation, thus enabling the retention and spread of poly-GA aggregates via the endocytic–exosomal pathway [67]. Although the mechanisms of cell-to-cell transmission may vary, all DPRs have indeed been observed to spread between cells in vitro [34,68,69,70]. Thus, it will be interesting to study (in the future) whether the endocytic–exosomal pathway represents a predominant mechanism of cell-to-cell transmission in C9-ALS/FTD (Figure 1D).

## 3. DPRs as Modulators of Stress Responses

In addition to activating the UPS, autophagy, endolysosomal, and secretory pathways, one additional way that neurons are able to respond to proteotoxic stress elicited by DPRs is by triggering intracellular stress response mechanisms that eventually repress translation and rebalance protein homeostasis. As it is the case of protein clearance pathways, an increasing body of evidence indicates that C9-associated DPRs both activate as well as disrupt stress pathways in multiple ways (Figure 2). The modulation of stress further reinforces disease pathology as described below.

### 3.1. DPRs Induce Chronic ER stress

A major part of cell stress responses occurs at the endoplasmic reticulum (ER) and results in the activation of the unfolded protein response (UPR), which eventually suppresses global translation and induces catabolic pathways, such as autophagy and ER associated protein degradation (Figure 2A). While acute ER stress can thus help to maintain protein homeostasis, sustained ER stress is well established to be cytotoxic and is in fact a hallmark of different forms of ALS [71]. Indeed, UPR markers are upregulated in distinct brain regions of patients with C9 ALS/FTD [72,73]. In the cerebellum in particular, the upregulation of phosphorylation of PERK, eIF2α, a regulatory subunit of the eukaryotic initiation factor 2 (eIF2), and IRE1α, all of which are major regulators of ER stress, is associated with the presence of poly-GA aggregates (Figure 2B) [73]. In line with this, the overexpression of poly-GA has been demonstrated to induce ER stress in cell culture [13,74]. Recent studies have further demonstrated that poly-PR can also promote ER stress in cell culture [75,76]. Importantly, pharmacological inhibition of ER stress suppresses DPR-induced neurotoxicity, suggesting that ER stress might represent a major aspect of pathogenesis [13,77,78,79].

Interestingly, while phosphorylation of eIF2α is required to inhibit canonical translation upon the activation of ER stress, a number of studies have demonstrated that p-eIF2α enhances the RAN translation of G4C2 transcripts [80,81,82,83,84] as well as other repeat RNAs [85]. Indeed, the RAN translation of myotonic dystrophy type 2-associated repeat transcripts was shown to be reduced in PERK knockout cells [85]. While the exact mechanisms underlying RAN translation remain unclear, these studies point toward a pathogenic feed-forward loop, whereby DPRs enhance their own production via the ER stress response (Figure 2C). Interestingly, C9orf72 has been demonstrated to enhance the interaction between eIF2α and eIF2B5, an eIF2-specific GEF. As a result, *C9orf72* knockout per se leads to global translation inhibition in vivo [86]. Future work will be required to assess whether disease-associated *C9orf72* haploinsufficiency could also contribute to an enhanced RAN translation of G4C2 transcripts. A number of studies have demonstrated that DPR toxicity can also suppress global translation by interacting directly with ribosomal components (Figure 2D) [87,88,89]. In particular, poly-GR/PR has been shown to bind to and block the polypeptide exit site of the ribosome, leading to a block in translation [89].

### 3.2. DPRs Disrupt Stress Granule Homeostasis

Another cellular response to proteotoxic stress is the formation of cytoplasmic stress granules (SGs). This subgroup of ribonucleoprotein granules are membrane-less organelles that form in the cytoplasm concomitant to global translational suppression to sequester away non-translating mRNA, translation initiation complexes, and RNA binding proteins (RBPs) [90,91]. Consistent with this, ER stress is an established inducer of SG formation [92]. In line with the role of DPRs in inducing ER stress, it is therefore not surprising that the overexpression of DPRs per se appears to increase spontaneous formation of SGs [14,78,80,83,93,94,95,96,97]. Indeed, DPR-induced SG formation has been shown to be dependent on the phosphorylation of eIF2α [95], as well as by the activation of the c-Jun N-terminal kinase (JNK) [79]. Taken together, DPRs thus appear to indirectly increase SG formation through ER stress signaling cascades (Figure 2E).

Beyond the indirect involvement in SG formation, emerging evidence suggests that DPRs can directly interact with SGs to change their properties and dynamics [78]. Physiological SG formation relies on liquid–liquid phase separation (LLPS), a type of phase transition whereby macromolecules can reversibly and spontaneously separate from the soluble phase (e.g., the cytoplasm) into a concentrated, condensed state [98]. This is driven in part by proteins with low-complexity domains (LCDs), composed of a high number of uncharged polar amino acids, which are found in many RBPs. Intriguingly, the involvement of LCDs in driving LLPS also points toward an involvement of the arginine- and glycine-rich DPRs, poly-PR, and poly-GR in SG formation. Indeed, poly-PR has been shown to undergo LLPS in vitro [95], and the interactomes of both DPRs are enriched in common SG components, including the RBPs hnRNPA1, TIA1, and FUS [78,87,93,94,99]. Importantly, the arginine-containing DPRs appear to reduce the liquid-like properties of hnRNPA1, TIA1, and FUS liquid droplets in vitro [93,95]. This may, at least in part, be due to increased beta sheet content of the liquid droplets, making them less dynamic [95]. Taken together, poly-GR/PR may thus contribute to disease pathogenesis by promoting the transition of SGs into a more solid-like state (Figure 2F). Indeed, a number of ALS associated mutations in the LCDs of RBPs have been linked to perturbed SG dynamics [100], suggesting that this may be a common mechanism underlying ALS disease pathogenesis.

Finally, emerging evidence suggests that DPRs may hijack SG formation to promote their own aggregation. In particular, knocking out G3BP1/2, a core driver of SG formation [101,102,103], leads to a virtual abolishment of cytoplasmic poly-GR inclusions [104]. Consistent with this, G3BP1/2 knockdown suppresses G4C2 toxicity in vivo [93]. The converse has also been shown by the overexpression of SG genes [99]. Further work will need to be carried out to decipher the relationship between the LLPS-mediated formation of SGs and protein aggregates, although several studies have shown that a liquid phase can precede the formation of solid protein aggregates of several ALS- and FTD-associated proteins [105,106,107,108,109].

## 4. Open Questions

Just over 10 years since the discovery of the G4C2 expansion in a non-coding region of the *C9orf72* gene, a myriad of disease mechanisms have been proposed to explain the effects of *C9orf72*’s loss of function as well as of the toxicity associated with repeat RNA and DPR production. However, a clear understanding of the relative importance of each disease mechanism, necessary to identify vulnerabilities that could be amenable to pharmacologic and medical interventions, is dramatically lacking. ALS/FTD, associated with *C9orf72* G4C2 expansion, remains an incurable disease and progress is marred in no small part by the pleiotropism of DPRs effects.

In this review, we have presented accumulating evidence revealing that multifaceted disruptions of protein homeostasis could be a major trait associated with DPR toxicity. Specifically, DPR toxicity appears to cause the inhibition of protein clearance pathways as well as the induction of chronic ER stress and aberrant dynamics of SG. While correlative studies were obtained from a wealth of in vitro, cellular, and in vivo model systems capable of investigating multiple aspects of ALS/FTD, the relevance of DPR-induced disruptions in protein homeostasis to pathogenesis remains poorly understood.

In the following, closing paragraphs, we discuss what we believe to be the key aspects that may help to untangle the intricacies of these pleiotropic DPR effects, moving the field toward the establishment of cause–effect relationships with specific relevance to disease pathology.

### 4.1. Which DPR Is the Most Toxic in Patients?

Several discrepancies regarding the contribution of DPR toxicity to disease progression become apparent when comparing model systems and tissues from patients suffering from ALS/FTD with the *C9orf72* G4C2 expansion (Table 1). In particular, the overexpression of the arginine-containing DPRs, but not poly-GA or poly-PA, appears to induce degeneration in the *Drosophila* eye [20,110]. In contrast, the overexpression of either poly-GA [111] or poly-GR [54] in mice causes neurodegenerative phenotypes, while poly-PR overexpression leads to notably milder and low penetrance phenotypes, with 60% of mice displaying no neurogenerative phenotypes by the age of 14 months [112]. Importantly, despite extensive post-mortem analysis of patient tissue, most studies have failed to demonstrate a clear spatial correlation between the presence of DPR inclusions and neurodegeneration [30,31,40,113,114]. In fact, DPR-positive inclusions appear to be quite rare, at least in comparison to the number of TDP43-positive inclusions found within the brains of patients with C9 ALS/FTD [41]. Intriguingly, one recent study has demonstrated a moderate correlation between the density of poly-GR aggregates and the extent of neurodegeneration in the frontal cortex [115], where the cytoplasmic poly-GR aggregates appear to be relatively abundant [116]. While this appears to be in line with the observation that poly-GR is toxic in *Drosophila* and mice, it remains inherently difficult to interpret the functional relevance from correlative observations across model systems.

### 4.2. How Do DPRs Induce Toxicity in Physiologically Relevant Conditions?

The level of DPR expression used in cell culture and within in vivo model systems, which largely rely on G4C2 overexpression or the treatment with synthetic DPRs, are likely much higher than in patients. Indeed, DPRs are expressed at relatively low levels in patient-derived iPSCs. This highlights the need to validate the reported effects of DPR toxicity in patient tissues and patient-derived cells, in which the toxicity of endogenous levels of DPRs can be specifically investigated. Along these lines, it may also be important to consider a recent study showing that common protein tags can affect the toxicity of DPRs in vivo [117], again emphasizing the importance of studying endogenous DPRs.

In recent years, a lot of effort has been made to investigate the specific toxicity of each individual DPR. While this has revealed both some overlapping and some distinct pathways that appear to be targeted by each DPR (Table 1), these studies leave an important question unanswered: how might the interaction between DPRs contribute to toxicity? Initial evidence suggests, for example, that the co-expression of poly-PA and GA in the chick embryonic spinal cord can reduce poly-GA aggregation and toxicity [118]. Conversely, in another co-expression study, poly-GA was found to promote the aggregation and thereby partially reduce the toxicity of poly-GR [119]. Again, whether such functional interactions also occur between endogenous DPRs under physiological conditions will need to be explored. Along these lines, it may be important to consider that DPRs generated from the sense strand (poly-GA, GP, and GR) appear to be relatively more abundant in patients with C9 ALS/FTD than those formed only by the antisense strand (poly-PA and poly-PR) (Table 1) [113]. Again, how this ratio may affect the toxicity of DPRs will need to be investigated thoroughly.

### 4.3. Why Is Each DPR Toxic?

The conformation of each DPR might widely affect toxicity as implied by the fact that the aggregation status can alter DPR toxicity. Considering their differing biochemical properties, it is likely that the toxic form will be different for each DPR. The hydrophobic and uncharged poly-GA, for example, is highly aggregation-prone and indeed appears to induce toxicity by sequestering components, such as the UPS, into their aggregates. Strikingly, disruption of the aggregation propensity via the insertion of proline residues completely abolished GA toxicity in primary neuronal cell culture [35], suggesting that poly-GA is most toxic in its aggregated form. Whether the positively charged poly-GR, for example, is more toxic in its soluble form could provide important improvements to understanding the mechanisms underlying DPR toxicity. These considerations have particular importance when evaluating how DPRs are cleared by protein degradation and how their disruptions of protein homeostasis may contribute to disease progression. The possible DPR-induced reduction in autophagic clearance, for example, may not affect the toxicity of soluble DPRs but may be a critical enhancer of the aggregation-prone poly-GA.

## 5. Conclusions

While the correlation between DPR toxicity and disruptions in protein homeostasis are evident, cause–effect relationships are currently hard to derive from the large body of data obtained in different systems displaying DPR accumulation. For example, while ER stress appears to be a hallmark of DPR toxicity, it is unclear how DPR toxicity may induce ER stress and whether this also contributes to DPR-induced disease progression.

Some insight may be gained from studying the interdependencies of protein homeostasis pathways (Figure 3). Specifically, while these pathways normally act in close concert to respond to proteotoxic stress, it is important to consider that crosstalk may instead perpetuate C9-ALS/FTD disease progression. Proteasome inhibition, for example, is a well-established inducer of ER stress [120,121], which leads to the suppression of global translation to reduce the burden of protein accumulation. In the context of DPR toxicity, however, another outcome emerges, as ER stress promotes the RAN translation of G4C2 transcripts and therefore paradoxically enhances the burden of toxic DPR accumulation [80,81,82,83,84]. It is therefore conceivable that the inhibition of the UPS may not only reduce the clearance of DPRs but may also indirectly increase their translation (Figure 3A).

ER stress induction and subsequent translational suppression also leads to the formation of SGs, which sequester non-translating mRNA, translation initiation complexes, and RBPs. Indeed, proteasome inhibition has been linked to increased SG formation [108,122,123]. DPRs, however, appear to trigger the persistence of SGs by reducing their liquid-like properties and may even use SGs to mediate their own aggregation. Again, the burden of toxic DPR accumulation is thereby perpetuated. With increased DPR translation and aggregation, it is conceivable that proteasome activity may be further inhibited, ER stress may be chronically induced, and aberrant SG formation may be further enhanced. Taken together, this reveals a self-sustaining feedback loop that perpetuates DPR toxicity (Figure 3B).

Along these lines, it is also interesting to consider the crosstalk between autophagy, stress granules, and the NPC. Importantly, autophagy is emerging as a mediator of SG clearance [108,124,125]. Could the DPR-induced impairments in autophagy therefore also contribute to the persistence of aberrant SGs? Moreover, the persistence of SGs may conversely reduce the TFEB-mediated transcriptional activation of the autophagy pathway, via NPC disruptions. Specifically, DPR-induced SGs have been shown to sequester components of the NPC, including the nucleoporin POM121 [54,126], which is a known modulator of TFEB-mediated autophagy [49]. As a result, DPRs and SGs further accumulate in a vicious cycle (Figure 3C).

Ultimately, while the crosstalk between catabolic and anabolic pathways is essential for maintaining protein homeostasis, their interplay appears to create self-sustaining pathogenic loops, potentially amplifying the pathogenic effects of the DPRs. Deciphering the contributions of feedback and feedforward interdependencies will be crucial to identifying key molecular targets to develop future therapies.

## Figures and Tables

**Figure 1 cells-13-00178-f001:**
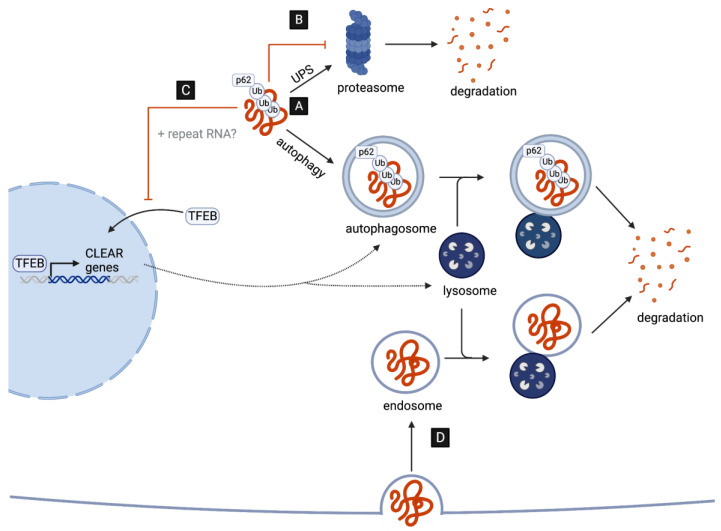
DPRs as targets and disruptors of cellular clearance pathways. (**A**) C9-associated DPRs are targeted for degradation by the ubiquitin–proteasome system (UPS) and the autophagy pathway. (**B**) DPRs impair the UPS by directly interacting with and inhibiting the 26S proteasome. (**C**) DPRs (and repeat RNAs) disrupt the autophagy pathway reducing the nuclear translocation of TFEB, the master transcriptional regulator of autophagy genes (CLEAR network). (**D**) G4C2-induced impairment in lysosome function may aid the cell-to-cell transmission of DPRs via endocytosis.

**Figure 2 cells-13-00178-f002:**
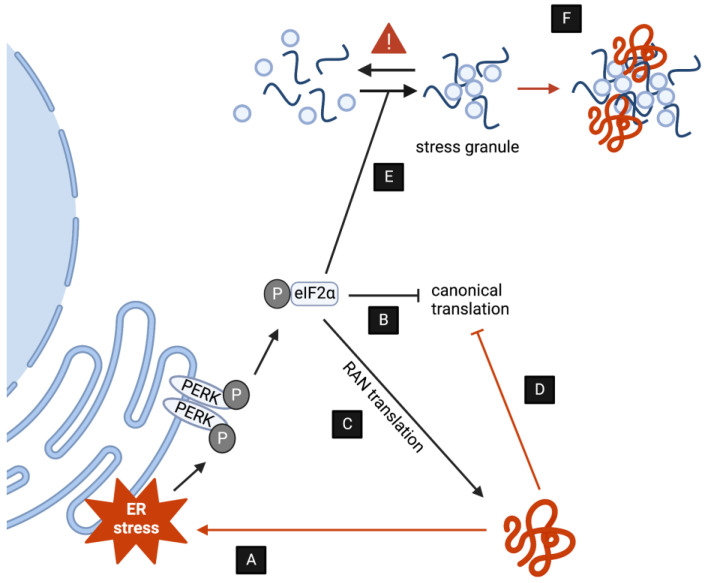
DPRs induce toxicity by upsetting stress responses. (**A,B**) DPR accumulation induces ER stress via PERK and other kinases, ultimately decreasing translation as a compensatory mechanism. (**C**) The induction of ER stress by DPRs promotes RAN translation. (**D**) DPRs interact directly with the ribosome blocking polypeptide formation. (**E**) DPRs indirectly stimulate the formation of SGs. (**F**) (some) DPRs are subjected to LLPS transitions and might directly alter SG dynamics.

**Figure 3 cells-13-00178-f003:**
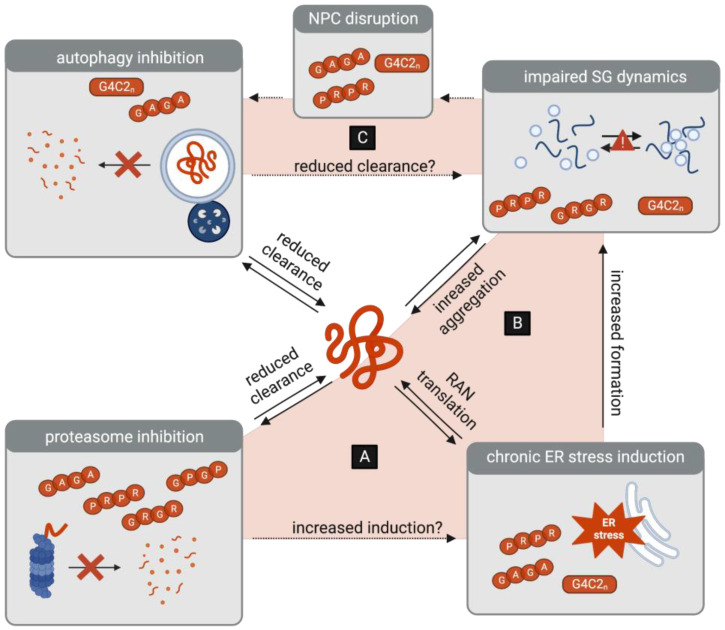
Interplay between protein homeostasis pathways as a potential amplifier of DPR toxicity. (**A**) DPR-associated inhibition of the proteasome may be a trigger of ER stress, which in turn leads to the increased RAN translation of G4C2 transcripts. (**B**) DPR-induced ER stress leads to the formation of stress granules (SGs), which in turn may lead to the increased aggregation of DPRs. (**C**) DPR-associated inhibition of autophagy may lead to the reduced clearance of SGs. Conversely, DPR-associated SGs contribute to the disruptions in the nuclear pore complex (NPC), which may further inhibit the initiation of autophagy.

**Table 1 cells-13-00178-t001:** Evidence of DPR toxicity from the tissues of patients with C9 ALS/FTD and in vivo model systems. Relative abundance of each DPR in patient tissue is indicated by “+” (++++ > +). DPR = dipeptide repeat, ND = neurodegeneration, UPS = ubiquitin–proteasome system, SGs = stress granules.

DPR	C9-Patient Tissue		Models Systems	
	Rel. Abundance	Correlation with ND	In Vivo Toxicity	Proceses Affected
Poly-GA	++++	no evidence	mouse (CFP-GA_149_, [111]; GFP-GA_175_, [112])	UPS, autophagy, ER stress
Poly-GP	+++	no evidence	no evidence	UPS
Poly-GR	++	yes	*Drosophila* eye and adult neurons (GR_36_ and GR_100_, [20]), mouse (GFP-GR_200_, [54])	UPS, translation, SGs
Poly-PR	+	no evidence	*Drosophila* eye and adult neurons (PR_36_ and PR_100_, [20]; PR_50_, [110]), mouse (GFP-PR_175_, [112])	UPS, translation, ER stress, SGs
Poly-PA	+	no evidence	no evidence	N/A

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
