# Peer review of "ALS’ Perfect Storm: C9orf72-Associated Toxic Dipeptide Repeats as Potential Multipotent Disruptors of Protein Homeostasis"

_cells, 2024, doi:10.3390/cells13020178_

Round 1

Reviewer 1 Report

Comments and Suggestions for Authors

In this review, the authors focussed on the effect of dipeptide repeats (DPRs), specifically in the C9orf72 gene as observed in amyotrophic lateral sclerosis (ALS) and frontotemporal dementia (FTD), on protein homeostasis pathways.

The aim of this review is straightforward and extremely relevant for various reasons. Disruption in protein homeostasis is a common hallmark of many late-onset protein misfolding diseases. However; not much is known about how a specific repeat expansion targets protein degradation pathways and impacts stress response machinery to alter protein homeostasis. Here, the authors attempted to bridge this gap and thus in my opinion, this review is going to be a valuable addition to the field.

 There is a growing interest in the proteostasis field regarding stress granules -- it's composition, function and regulation in the context of pathogenesis associated with protein misfolding. Here, the authors discussed how DPRs increase stress granule formation through ER stress-signaling  cascades. Finally, the authors discussed how DPRs are thought to hijack stress granule formation to promote their own aggregation. All these contribute to alteration in stress granule homeostasis that plays an important role in protein misfolding-induced toxicity. In this review, the authors discussed this new and emerging topic in great detail which is highly appreciable.

This review is very comprehensive and covers all the important articles published in this field. I appreciate how the authors concluded with the open questions which reflects the authors' insight on this topic. 

All the references are appropriate and relevant. No specific comments for the tables and figures. They look alright and can be accepted for publication in their present form. 

Author Response

We thank the reviewer for her/his words of praise and support of our review. We are happy that she/he finds that the review "can be accepted for publication in their present form".

Reviewer 2 Report

Comments and Suggestions for Authors

This review manuscript is comprehensive, focused and inspiring. To me only some minor issues need to be addressed.

Minor issues:

Page 1, line 7: Please change “correct balance” to “balanced regulation”.

Page 1, line 29-30: Please change “decline” to “dysregulation”.

Page 1, line 35-36: This sentence is an overstatement. Please change “both diseases are caused by” to “a good portion of patients of both diseases have been found to carry”.

Page 2, line 50: Please change “repeat RNA” to “repeat RNAs”.

Page 2, line 55: Please change “repeat RNA can be translated by” to “repeat RNAs can go through”.

Page 2, line 88: “in vivo” also includes “in the brain”. Do the authors mean “in animal models”?

Page 2, line 93: Please change “adapt” to “stuck in”.

Page 3, line 106: Please change “initial” to “primitive”.

Page 3, line 109-110: Please elaborate on how DPRs inhibit proteasome activities in an “non-cell-autonomous” manner. If the authors refer to the exocytosis-endocytosis process, please make it clearer. Please also change “begin to” to “help”.

Page 3, line 125: Please delete “a” before “low pH”.

Page3, line 142: Please change “repeat RNA” to “repeat RNAs”.

Page 4, line 152: Please add “the” before “presence of”. Please change “repeat RNA” to “repeat RNAs”.

Page 4, line 164: Please change “also C9orf72-deficient cells” to “C9orf72-deficient cells also”.

Page 5, line 188: Please change “repeat RNA” to “repeat RNAs” in the figure legend and inside Figure 1.

Page 6, line 226: Please change “repeat RNA” to “repeat RNAs”.

Page 6, line 227: Please delete the comma before “was shown”.

Page 6, line 238: Please add a comma between “ribosome” and “leading”.

Page 8, line 290: Pleas change “ER Scheme2” to “ER stress”.

Page 10, line 362: Please add a comma after “Conversely”.

Page 11, line 391: Please add “different” before “systems”.

Page 11, line 409: Please add comma before and after “however”.

Page 12, Figure 3: Please correct the typo “increased aggregation” to “increased aggregation”.

Author Response

This review manuscript is comprehensive, focused and inspiring. To me only some minor issues need to be addressed.

A: We thank the reviewer for her/his very thorough proofreading of our review. We are happy that she/he finds that “only minor issues need to be addressed”.

Minor issues:

Page 1, line 7: Please change “correct balance” to “balanced regulation”.

A: we corrected as suggested

Page 1, line 29-30: Please change “decline” to “dysregulation”.

A: we corrected as suggested

Page 1, line 35-36: This sentence is an overstatement. Please change “both diseases are caused by” to “a good portion of patients of both diseases have been found to carry”.

A: we corrected as suggested

Page 2, line 50: Please change “repeat RNA” to “repeat RNAs”.

A: we corrected as suggested

Page 2, line 55: Please change “repeat RNA can be translated by” to “repeat RNAs can go through”.

A: we corrected as suggested

Page 2, line 88: “in vivo” also includes “in the brain”. Do the authors mean “in animal models”?

A: we corrected as suggested

Page 2, line 93: Please change “adapt” to “stuck in”.

A: we corrected as suggested

Page 3, line 106: Please change “initial” to “primitive”.

A: we corrected as suggested

Page 3, line 109-110: Please elaborate on how DPRs inhibit proteasome activities in an “non-cell-autonomous” manner. If the authors refer to the exocytosis-endocytosis process, please make it clearer. 

A: we now provide more specific details about the study discussed (see added sentence starting at line 145 in the marked-up revised version) 

Please also change “begin to” to “help”.

A: we corrected as suggested 

Page 3, line 125: Please delete “a” before “low pH”.

A: we corrected as suggested 

Page3, line 142: Please change “repeat RNA” to “repeat RNAs”.

A: we corrected as suggested 

Page 4, line 152: Please add “the” before “presence of”. Please change “repeat RNA” to “repeat RNAs”.

A: we corrected as suggested 

Page 4, line 164: Please change “also C9orf72-deficient cells” to “C9orf72-deficient cells also”.

A: we corrected as suggested 

Page 5, line 188: Please change “repeat RNA” to “repeat RNAs” in the figure legend and inside Figure 1.

A: we corrected as suggested 

Page 6, line 226: Please change “repeat RNA” to “repeat RNAs”.

A: we corrected as suggested 

Page 6, line 227: Please delete the comma before “was shown”.

A: we corrected as suggested 

Page 6, line 238: Please add a comma between “ribosome” and “leading”.

A: we corrected as suggested 

Page 8, line 290: Please change “ER Scheme2” to “ER stress”.

A: we corrected as suggested 

Page 10, line 362: Please add a comma after “Conversely”.

A: we corrected as suggested 

Page 11, line 391: Please add “different” before “systems”.

A: we corrected as suggested 

Page 11, line 409: Please add comma before and after “however”.

A: we corrected as suggested 

Page 12, Figure 3: Please correct the typo “increased aggregation” to “increased aggregation”.

A: we corrected as suggested

Reviewer 3 Report

Comments and Suggestions for Authors

In this study authors propose a disease mechanism of C9-FTD-ALS in which DPR aggregates disrupts autophagy, UPS and ER stress. This interaction was already described, and the novelty is debatable. However, compiling the current evidence and the models are appreciated. Authors propose DPR pathology as a disease driver mechanism but the evidence in human pathology, as they also say, is very weak (line 329-331). Interestingly, results employing ASOs targeting C9orf72/DPR expression (BIIB078) clinical trial did not report any benefits, weakening the translation of this review. Alternatively, author should comment and include this important information from a clinical trial. There is a lack of discussion of human pathology and the review is mainly based in artificial overexpression models that produces a higher amount of DPRs. In this line, authors should state the cell model or in vitro system employed when cited in the text (in line 134, for example). It is highly relevant when talking about NPC because its turnover in post-mitotic cells is very low, in contrast with dividing cells commonly used in cell cultures.

There are also minor issues that should be amended:

1- In line 38-39 authors explain the importance of G4C2 repeats and set the limit in above 100. Nevertheless, in Iacongeli et al., 2019 meta-analysis an association with 24-30 repeats is described. Authors should include conflicting results in this topic because is relevant for DPR formation.

2- In line 101-102 authors say that TDP-43 is an “aggregation-prone protein associated with familiar forms of ALS.” Not only in familiar forms but also in sporadic forms. In fact, it accounts for the 97% of ALS patients.

3- Reference Bozič et al., 2020 does not exist and it should be Bozič et al., 2022.

4- Table 1 should contain more information such as a column with the selected references and the exact model system/cell culture employed (catalogue number of the mouse model, promoter and G4C2 number of copies in the case of Drosophila…). An explanation for “weak” evidence would make the table more comprehensible.

Author Response

In this study authors propose a disease mechanism of C9-FTD-ALS in which DPR aggregates disrupts autophagy, UPS and ER stress. This interaction was already described, and the novelty is debatable. However, compiling the current evidence and the models are appreciated. 

A: We thank the reviewer for appreciating our discussion of the current literature.

Authors propose DPR pathology as a disease driver mechanism but the evidence in human pathology, as they also say, is very weak (line 329-331). Interestingly, results employing ASOs targeting C9orf72/DPR expression (BIIB078) clinical trial did not report any benefits, weakening the translation of this review. Alternatively, author should comment and include this important information from a clinical trial. 

A: We concur with the reviewer and added the information as suggested (see added sentences starting at line 55 in the marked-up revised version). We hope that the reviewer will now feel that we provide a more balanced view reflecting that essentially it is still unknown whether and to what extent toxicity is linked to DPRs or loss of C9ORF72 function. 

 There is a lack of discussion of human pathology and the review is mainly based in artificial overexpression models that produces a higher amount of DPRs. In this line, authors should state the cell model or in vitro system employed when cited in the text (in line 134, for example). It is highly relevant when talking about NPC because its turnover in post-mitotic cells is very low, in contrast with dividing cells commonly used in cell cultures.

A: We now added more details about the discussed experiments (see modifications at lines 187, 190 and 192 in the marked-up revised version). 

There are also minor issues that should be amended:

1- In line 38-39 authors explain the importance of G4C2 repeats and set the limit in above 100. Nevertheless, in Iacongeli et al., 2019 meta-analysis an association with 24-30 repeats is described. Authors should include conflicting results in this topic because is relevant for DPR formation.

A: We agree with the reviewer and added the information as suggested (see added sentences starting at line 39 in the marked-up revised version).

2- In line 101-102 authors say that TDP-43 is an “aggregation-prone protein associated with familiar forms of ALS.” Not only in familiar forms but also in sporadic forms. In fact, it accounts for the 97% of ALS patients.

A: We thank the reviewer for identifying this oversight. We correct the information as suggested (see modified sentence at lines 136-137 in the marked-up revised version).

3- Reference Bozič et al., 2020 does not exist and it should be Bozič et al., 2022.

A: We corrected as suggested.

4- Table 1 should contain more information such as a column with the selected references and the exact model system/cell culture employed (catalogue number of the mouse model, promoter and G4C2 number of copies in the case of Drosophila…). An explanation for “weak” evidence would make the table more comprehensible.

A: We thank the reviewer for the suggested improvements. We have added the selected references and repeat numbers for mice and Drosophila models in the third column of the revised table. Also, we now comment about the phenotype penetrance in the text (see modified line 384-386 in the marked-up revised version).

Round 2

Reviewer 3 Report

Comments and Suggestions for Authors

In this new version authors succesfully corrected the manuscript based on the previous review report.